# The Leg Sensilla of Insects from Different Habitats—Comparison of Strictly Aquatic and Riparian Bugs (Corixidae, Ochteridae, Gelastocoridae: Nepomorpha: Insecta: Heteroptera)

**DOI:** 10.3390/insects14050441

**Published:** 2023-05-07

**Authors:** Agnieszka Nowińska, Barbara Franielczyk-Pietyra, Dan A. Polhemus

**Affiliations:** 1Faculty of Natural Science, Institute of Biology, Biotechnology and Environmental Protection, University of Silesia in Katowice, Bankowa 9, 40-007 Katowice, Poland; agnieszka.nowinska@us.edu.pl; 2Department of Natural Sciences, Bishop Museum, 1525 Bernice St., Honolulu, HI 96817, USA; bugman@bishopmuseum.org

**Keywords:** Corixidae, Gelastocoridae, Ochteridae, Nepomorpha, legs, sensilla

## Abstract

**Simple Summary:**

Insects of the Nepomorpha infraorder live in various environments—some are associated strictly with water and some are mostly terrestrial. In this study, we examined specimens from the aquatic Corixidae and two riparian families—Ochteridae and Gelastocoridae. Apart from the differences in general leg morphology (legs adapted for swimming in Corixidae and for walking in Ochteroidea), there were eight types of sensilla found on the legs. From those, mechanosensilla displayed the most observed variability. The leg sensilla of Nepomorpha had not been studied until now.

**Abstract:**

The aim of this study was to analyze morphological types and arrangement of the leg sensilla of Corixidae, Ochteridae and Gelastocoridae, in relation to their various habitats. The leg sensilla of four species of Corixidae, six of Gelastocoridae and two of Ochteridae were studied. Eight main types of sensilla with six subtypes of sensilla trichodea and four subtypes of sensilla chaetica were found and described. The greatest variability was observed among mechanoreceptive sensilla. The study showed differences in the shape of the legs between strictly aquatic and terrestrial taxa. It is the first attempt to describe leg sensilla among nepomorphan taxa.

## 1. Introduction

The infraorder Nepomorpha (true water bugs) comprises six superfamilies—Naucoroidea, Notonectoidea, Pleoidea, Nepoidea, Ochteroidea and Corixoidea [1]. The first phylogenetic studies concerning Corixoidea distinguished only the family Corixidae, as one of 11 families among Nepomorpha [2]. Following the analysis of Nieser [3], two other taxa—Micronectidae and Diaprepocoridae—were upgraded to a family level. The majority of other scientists supported this change [4,5,6,7,8]. The systematic position of Corixoidea has also been studied by many authors [1,9,10,11,12,13,14,15,16] and different results were obtained. However, the majority of the authors, including the most recent results of Ye et al. [1], place Corixoidea as a second branch of the phylogenetic tree after the clade Nepidae + Belostomatidae. The results of the above-mentioned authors [1] also suggest that Corixoidea, containing one family, Corixidae, with 35 genera [4], is the sister group to all other Nepomorpha except the superfamily Nepoidea.

Corixidae, known also as water boatmen, are distributed worldwide, with particular species richness in northern temperate zones. They have been recorded from a wide array of habitats, including both fresh and saline, still and running, etc., and even from acidified water. Their diet is varied, from algae to dead arthropods, which means they are omnivores [17,18,19]. They are also susceptible to predation by fish, with studies indicating that an absence of fish results in an increase in corixid density [17]. Because of their oarlike hind legs, they are good swimmers, moving with their thorax up. The body length is between 2.5 and 15 mm, and the tarsal formula is 1-1-2 [4].

The second studied superfamily, Ochteroidea, has been placed on different branches within the phylogeny of Nepomorpha by various authors [1,12,13], and therefore its exact relationships are still uncertain. However, in each of these studies it is found to consist of two families—Ochteridae (with three genera: *Ochterus, Megochterus,* and *Ocyochterus*) and Gelastocoridae (with two genera: *Gelastocoris* and *Nerthra*) [20]. In the most recent phylogenetic study, Ye et al. [1] also suggested that at least six independent transitions from lentic to other habitats occurred during the evolution of some nepomorphans. One of these transitions is hypothesized to have occurred in Ochteroidea—a transition from lentic to riparian habitats.

Ochteroidea are known for their secondarily terrestrial lifestyle, and for their ability to camouflage against riparian substrates. Gelastocoridae (toad bugs) are larger insects (7–15 mm) resembling toads (able to jump), are often mud-covered, and have a tarsal formula 1-2-3. On the other hand, Ochteridae (velvety shore bugs) are smaller (4.5–9 mm), often possess a black ground color with small yellow or lavender spots and have the tarsal formula 2-2-3. Representatives of both families do not have legs specially adapted for swimming (due to a lack of swimming hairs), but rather for walking [20,21].

The legs of Heteroptera consist of five elements: the coxa, trochanter, femur, tibia and tarsus, the latter bearing the claws. The appearance of some segments varies between groups in Nepomorpha. Often, the femur of the foreleg is strongly developed. It is usually used to trap and hold prey (between a bent femur, tibia and tarsus), representing a “raptorial leg” [4,21]. Among the above-mentioned families, only Gelastocoridae have raptorial forelegs. Other unique foreleg modifications are present in species of Corixidae: the fore tarsus is called the “pala” and the arrangement of setae and their size is a useful diagnostic feature for separating genera and species [4]. The pala takes the form of a spoon-like element bearing a claw and long bristles. Its ventral side has short, thick seta, called pegs. Males of the majority of Corixidae subfamilies have the pegs in a row, with their size and arrangement representing useful identification features [4]. The middle and hind tarsi of Corixidae usually have a pair of long claws.

Apart from walking, nepomorphan legs can also be used for stridulation. Among others, Corixidae (of both sexes) have forelegs designed to encourage the female during mating [4].

On the surface of the insects’ legs multiple sensory structures (sensilla) are present, mainly of mechano- and chemoreceptive function [22,23,24]. On the basis of external morphology, it is possible to determine the type of sensillum, based on the shape, surface (the presence and absence of pores) and socket (flexible or inflexible connection with the cuticle). This determination is possible thanks to the support of many ultrastructural studies on these structures (number of neurons, branched or non-branched dendrites, and sheath cells) [25,26,27,28,29,30].

Leg sensilla have not been previously studied in water bugs. In other insects, data on leg sensilla in general are sparse, and studies have mostly focused on chemosensilla and campaniform sensilla [31,32], particularly of the tarsi [33,34,35]. However, labial and antennal sensilla have already been described in both aquatic and semi-aquatic insects [15,36,37,38,39,40,41,42], becoming a good basis for comparison. A variety of sensilla types and subtypes were described. It has been stated that while semi-aquatic bugs (Gerromorpha) have a lot in common with terrestrial insects regarding antennal sensilla types, water bugs (Nepomorpha) differ greatly from terrestrial insects, especially in terms of diverse antennal mechanosensilla [36,37,38,39,40,41].

As a first attempt to investigate potential differences in the leg sensory equipment between strictly aquatic and riparian Nepomorpha, the external leg morphologies of selected specimens from strictly aquatic Corixidae, and riparian Ochteridae and Gelastocoridae, originating from museum collections, were studied. 

The research hypothesis is that, due to different environments, the leg sensilla of strictly aquatic bugs differ from the leg sensilla of riparian bugs in both morphology and number.

## 2. Materials and Methods

Materials were obtained from the collections of the Moravian Museum in Brno, the Natural History Museum in Prague and the Naturalis Biodiversity Center in Leiden (Netherlands) and are stored in the University of Silesia in Katowice, Institute of Biology, Biodiversity and Environmental Protection (Poland). The specimens obtained from the Naturalis Biodiversity Center in Leiden (Netherlands) are ultimately to be sent back and stored in the Naturalis Biodiversity Center. At least 2 individuals of each species were examined for this study; the sex of insects was not considered, as there were no notable intersexual differences in the sensilla types (based on preliminary studies) (Table 1). All specimens were cleaned in an ultrasonic cleaner (Polsonic, Warsaw, Poland), the legs were dissected, dried in ethanol, mounted, and then sputter-coated with gold with the use of the turbomolecular pump coater (Quorum 150 T ES plus—Quorum Technologies, Laughton, East Sussex, UK).

Both sides of the legs were observed with the use of the scanning electron microscope (Phenom XL Phenom-World BV the Netherlands) in the scanning microscopy laboratory of the Faculty of Natural Science, Institute of Biology, Biotechnology and Environmental Protection of the Silesian University in Katowice. We followed the terminology used in other papers on insects’ sensilla [26,29,36,37].

The legs of 12 species from three families were studied (Table 1).

## 3. Results

### 3.1. Gross Morphology of the First Pair of Legs

Because most of the morphological differences can be seen among the forelegs of the species studied, only these pairs of legs are compared here (Figure 1a–c).

In Corixidae (Figure 1a) and Ochteridae (Figure 1b), the coxa is elongated, and externally covered with hairs. On the dorsal side of the coxa an indentation is visible along 2/3 of its length, and in Corixidae (*Callicorixa, Sigara*), some hairs are present at the proximal part. On the other hand, in Gelastocoridae (Figure 1c) the coxa is also covered with hairs but the segment is shorter. There is a groove along the entire segment, which allows articulation of the tibia and tarsus while bending the leg. Representatives of all three families have a small grouping of thin hairs near the trochanter.

In regard to the femur, that of *Nerthra* (Gelastocoridae) is enlarged, and covered with brush-like hairs. *Gelastocoris* also has a broadened femur, with some thin hairs, and a distinctive group of hairs medially. The bottom part of the femur, from both ventral and dorsal sides, has two visible rows of thick pegs (cuticular appendages for better grasping during hunting). These pegs are also present on the femur of *Nerthra* species. *Ochterus* species (Ochteridae) have an elongate femur covered almost entirely with thin hairs. There is a structure resembling a comb on the dorsal side of the femur, but denser than that seen in *Gelastocoris* species, with three rows of thick, short hairs. Towards the end of the segment, the structure changes into loosely arranged hairs.

The most distinctive surface on the femur was observed among species of Corixidae. The dorsal surface of this segment has a triangular patch covered with hairs and circular areas without hair. The proximal part of the femur is also covered with the same sensilla as noted above on the coxa. Anterobasally, there are specialized stridulatory pegs: *pars stridens*. These structures are hair-like pegs that have a broadened base in males but are narrow in females. Beyond the indentation, the femur has a smooth surface with only a few hairs. The same segment on the ventral side is also partly covered with hairs, but also includes rounded fields without hairs.

The tibia in Corixidae is short, slightly bent and folded, and has a rather smooth surface. A few long, thin hairs grow out of this segment and reach the tarsus. In Ochteridae, the tibia is a long segment, covered with medium-length hairs, some of them flattened (closer ventral side), some thick and sharpened (closer dorsal side). Gelastocoridae species have a short tibia with a few hairs in *Nerthra*, while in *Gelastocoris* there are many hairs. Analyzing the ventral side of the tibia in *Gelastocoris*, there are some grouped hairs, probably to connect with hairs present at the dorsal side of the femur. Pegs are present, of the same type as seen on the femur in both genera, *Gelastocoris* and *Nerthra*.

The tarsus is most distinctive in the Corixidae. In this family, it is one-segmented and forms the pala. This spoon-like segment is covered with long hairs at its edges. The dorsal side has only a few long hairs and a rather smooth surface, while the ventral cavity/inner surface is full of shorter, thin hairs. The tarsus terminates in the pretarsus in the form of a sharpened claw, which has a serrated edge. The ventral side of the pala has various numbers and combinations of short pegs, which are characteristic features for male recognition. Gelastocoridae also possess a one-segmented, slim and elongated tarsus end in one claw, while in Ochteridae a two-segmented tarsus, with two claws, is present.

### 3.2. Categories and Morphology of Leg Sensilla

Eight main types of sensilla were found on the legs of species studied (Figure 2):Sensilla trichodea (**ST**)—long, thin, hairlike sensilla, with smooth or ribbed surfaces. They have a flexible socket (a thin membrane connects the cuticle of the leg with the cuticle of the sensillum, making it movable at the base), which gives them a putative mechanoreceptive function. The shape of this sensillum varies from tapered at the top to flattened at the top. The tip is either straight or bent. This type of sensilla appears in groups, covering large areas of the surface. The other function performed by sensilla trichodea is gustation. In this case, the sensillum occurs with a single pore on the tip. ST1—long, thin, hairlike sensilla with a ribbed surface, without pores—they perform a mechanoreceptive functionST2—long, thin, hairlike sensilla with a smooth surface, without pores—they perform a mechanoreceptive functionST3—long, flattened, sometimes curved inwards sensilla, more/less ribbed surface, without pores—they perform a mechanoreceptive functionST4—short flattened sensilla resembling a leaf with a frayed end, without pores—they perform a mechanoreceptive functionST5—long ribbed sensilla, flattening and widening along the length, with a ribbed frayed end resembling a brush, without pores—they perform a mechanoreceptive functionST6—long ribbed sensilla with frayed edges, without pores—they perform a mechanoreceptive functionSensilla chaetica (SCh)—thick sensilla with pronounced ribs on the surface. The length varies. The tip is either pointed or rounded. It has a flexible socket, like the sensilla trichodea, but is easy to distinguish from this other type because it is visibly thicker and more rigid. This type is also described in the literature as mechanoreceptive sensilla.
SCh1—thick and rigid sensilla with pronounced ribs on the surface. The sensillum is long, thicker at the base and tapers to the tip. The tip is sharpened or slightly rounded.SCh2—thick and rigid sensilla with pronounced ribs on the surface. The sensillum is short and slightly bent. Observed on the “pala”, arranged in rows.SCh3—thick and rigid sensilla with pronounced ribs on the surface. The sensillum narrows strongly towards the tip and ends with a thin, long tip.SCh4—thick and rigid sensilla with pronounced ribs on the surface with a clearly rounded tip and a well-developed socket.Sensilla campaniformia (SCa)—oval or elongated disks lying flat on the surface, usually with a visible pore in the middle. This type also has a flexible socket. These sensilla belong to mechanoreceptors and are described as pressure sensilla.Sensilla basiconica (SB)—cone-like structures, usually smaller than trichoid sensilla, with a porous or non-porous surface and inflexible socket (no membrane connecting the cuticle of the leg with the cuticle of the sensillum). In our studies, only the non-porous sensilla basiconica were observed. They are long, have a wrinkled surface, a round end, and occur in groups between the segments of the legs. They are believed to perform a proprioceptive function.Sensilla placodea multilobated (SPM)—round cavities with small, fingerlike protuberances. As they were observed before on the antennae of studied species, the name was given according to these other studies. The probable function is olfaction; however, olfactory structures are not specific for leg sensilla. Therefore, they might play another role.Sensilla coeloconica (SCo)—small cones with an inflexible socket and smooth surface. They have been observed as either single or covering a bigger part of the leg surface. They are believed to perform a thermo-hygroreceptive function.Sensilla ampullacea (SA)—peg in pit sensilla, with the opening being the only part visible on the surface. The body of the sensillum is hidden in a cavity and rises from an inflexible socket. The sensillum is believed to perform a thermo-hygroreceptive function.Sensilla styloconica (SS)—pegs arising from a bulge of cuticle, with a flexible socket and a ribbed surface. The sensillum is believed to perform mechanoreceptive or gustatory function.

### 3.3. Sensilla Observed among Studied Families

#### 3.3.1. Corixidae

Sensilla trichoidea ST1 and ST2 (Figure 2 and Figure 3c,d) were the only types of sensilla present on the coxae of each pair of legs and were generally most common in this group. Sensilla campaniformia were present on the trochanter of each leg, and also observed on the femur and tibia of the 1st and 2nd legs (Figure 2, Figure 3c and Figure 4a). Another type, sensilla chaetica (SCh1, SCh2, SCh3), of different lengths, were observed along the length of the leg, on the trochanter, femur, tibia and tarsus (Figure 2, Figure 3a,e,f and Figure 4b). Sensilla coelonica were noticed only on the femur of the foreleg (Figure 2 and Figure 4e). Another type of trichoid sensilla was present on the tibia of Corixidae, on the middle leg ST5 (Figure 2, Figure 3f and Figure 4c), while on the hindleg, ST6 (Figure 2 and Figure 3b) was found. The latter type was also noted on the tarsus of the hindleg. Additionally, other structures were observed, which might not be sensillar structures. On the femur, circular “gaps” in the hair pile with slightly raised edges (Figure 4d) were observed. Moreover, there was a distinctive type of structure photographed in *Sigara* species—flat, rounded fields with short, longitudinal indentations, on the femur and tibia of the first and second pairs of legs (Figure 4f). Apart from that, on the femur in both species of *Sigara*, there were pars stridens positioned anterobasally. *Calliocorixa* and *Heliocorixa* also had this element, but in smaller numbers.

#### 3.3.2. Gelastocoridae

Apart from ST1 and ST2 (Figure 2 and Figure 5d), the second most prominent sensillum in this group was SCh1, which was very common on leg segments among studied *Gelastocoris* species and also noticed on the trochanter of the foreleg in *Nerthra* species (Figure 2 and Figure 6c). Another sensillum, campaniformia, was noticed on the fore- and hindlegs (Figure 2 and Figure 5f). Among all the species studied, only in *Gelastocoris flavus* were sensilla basiconica present (Figure 2), occurring solely on the foreleg coxa. ST3 were also common, especially on the femur and tibia of all legs in *Gelastocoris* species and on almost all segments and legs of *Nerthra* species (Figure 2 and Figure 5a,b). Sensilla placodea multilobated (SPM) were seen only on the middle leg coxa of Gelastocoris species, but on almost all coxae and trochanters of all legs of *Nerthra* species (Figure 2 and Figure 5c). On the tarsus of the foreleg in both *Gelastocoris* species and two species of *Nerthra* (*N. ranina* and *N. mixta*), sensilla trichodea resembling contact chemoreceptors in plant feeders (ST1) were observed (Figure 2). Another type, sensilla coeloconica was found only on the tarsus of *Gelastocoris* species (Figure 2 and Figure 5e).

Distinctive in this group are single sensilla styloconica, found on the trochanter of *N. grandicolis* and *N. mixta* (Figure 2 and Figure 6c), and rows of sensilla chaetica (SCh4) observed on the edge of the femur and tibia of *Gelastocoris* (Figure 2 and Figure 5a). In the same species, sensilla trichodea type 4 (ST4) (Figure 2 and Figure 6a) were noticed on all legs. Only *G. flavus* had thermo-hygroreceptive sensilla ampullacea (Figure 2 and Figure 5d).

As for distinctive non-sensillar structures, in *N. colaticollis,* there were cuticular outgrowths with ribbed surfaces observed on the edge of the femur (Figure 6b).

#### 3.3.3. Ochteridae

In this taxon, as well as in Gelastocoridae, basiconic sensilla (SB) were observed on the coxae of the middle and hindleg of *Ochterus perbosci* (Figure 2 and Figure 7f) and on the femur of the foreleg of *O. marginatus* (Figure 2 and Figure 8a). ST1 and ST2 were the most common, while long ST1 were present on the tarsi of the 1st and 2nd pair, between the claws, as in Gelastocoridae (Figure 2 and Figure 7c). Another type of trichoid sensillum (ST3) was also present on the tibia of the middle and hindleg (Figure 2 and Figure 7c,e). Sensilla campaniformia were present on almost all segments of all three pairs of legs from this genus (Figure 2 and Figure 7b). Among studied *Ochterus* species, SPM were largely present in *O. marginatus* (Figure 2 and Figure 7d). Sensilla coeloconica were found only on the tibia of the middle leg (Figure 2 and Figure 7a,b). On the femur of each leg in this group, a peculiar hair grouping was found—sensilla trichodea (ST3), forming a comb-like structure (Figure 2 and Figure 8b).

## 4. Discussion

This paper presents the results of a morphological study of leg sensilla in 12 species in three families of Nepomorpha, conducted with a scanning electron microscope.

Until now, the sensillar structures of Nepomorpha were studied on the labium and antennae [15,37,38,39,40,41]. Leg sensilla have not been studied before in these taxa.

The biggest differences between the studied families can be drawn between taxa strictly associated with water (Corixidae), and taxa associated mostly with terrestrial riparian environments (Gelastocoridae and Ochteridae). First, the differences can be noted in the leg shapes. While the legs of Gelastocoridae and Ochteridae are adapted for walking, with thin long segments and tarsal claws, the legs of Corixidae are adapted for swimming. Their tarsi are flattened, shaped like oars, and the last pair of legs is elongated.

Additionally, the leg sensilla observed in Corixidae also differ from those of the other studied taxa.

Along the length of the leg segments, long, presumably mechanoreceptive sensilla with different types of fringes were observed (ST5 and ST6) (Table 2). Given their position and length, they probably assist with swimming by increasing the surface area of the flattened leg segments. Apart from these sensilla, most of the other mechanoreceptive sensilla (ST, SCh, SCa) were common to all the studied taxa, and are common on other body parts of these insects [39,42]. A more unique feature is the sensilla chaetica occurring on the foretarsus (pala) (SCh2) (Table 2). Along with the shape of the pala, they are diagnostic features for the species [4].

Potential mechanosensilla specific of the other two taxa were also described.

In Ochteoidea, long ST1 were present on the last tarsal segment. Presumably, in herbivorous insects, trichoid sensilla in this position function as gustatory sensilla, and their presence is common [26]. However, we did not observe a terminal pore on this sensillum, nor is the taxon herbivorous. The comparison is therefore strictly based on the external morphology of studied taxa and terrestrial insects. Some studies also suggest that ST might be responsible for pheromone detection [38]. The presence of these sensilla could confirm a strong connection to the terrestrial environment for Gelastocoridae and Ochteridae.

Another type found only in Gelastocoridae, sensilla styloconica (Table 2), are also found in specimens connected to the terrestrial environment. They were observed singularly in *Nerthra*.

A distinct type of sensilla was also observed in Ochteridae: basiconic sensilla with a putative proprioceptive function (Table 2). They form a group of long sensilla with a wrinkled surface, transferring the signals from the movement of two adjacent leg segments.

Presumed thermo-hygroreceptive sensilla were present in all studied taxa as a small number of mostly sensilla coeloconica (Table 2). Sensilla ampullacea were observed only in Gelastocoridae (Table 2). However, given their size and difficulty of recognition, it is possible that they are present in other studied taxa. Therefore, they do not play a significant role in our comparison. Especially given the fact that, potential thermo-hygroreceptive sensilla (SA and SCo) in general were observed in all the studied taxa (Table 2).

Some similarities can be found while comparing sensilla on different body parts. In Corixidae, hairlike sensilla trichodea as well as flat mechanoreceptive sensilla have been found on all legs, labia and antennae. On the labium, they were described as ribbon-like sensilla [42]; on the antennae, they are round at the base and flatten at the top (ST3) [39]; and on the legs, they are additionally frayed at the edges or at the ends. Other hair-like sensilla trichodea have also been described on all studied body parts, as well as sensilla coeloconica. Sensilla campaniformia were found on the antennae as well as on the legs. However, in contrast with other body parts, no porous sensilla were observed on the legs [39,42].

In Ochteridae, presumed mechanoreceptive sensilla are also present in the majority of the studied body parts. Besides hairlike sensilla, cupola and peg-like mechanosensilla were found on the labium. Proprioceptive sensilla were also found on all legs, labia and antennae. Additionally, as on the labia, no porous sensilla were observed on the legs of the studied specimens. Sensilla placodea multilobated (SPM) are the distinctive feature of the legs of Ochteridae. They were not observed on other studied body parts. However, they are present on the antennae in one other taxon—Gelastocoridae [15,38].

In Gelastocoridae, as mentioned above, the presence of SPM had already been noted on the antennae [38]. During these studies, they were also observed on the legs of Gelastocoridae, as well as in Ochteridae. Apart from SPM, as in other taxa, mechanoreceptive sensilla constituted the majority of sensilla present on all studied body parts. On the labia, tactile/gustatory sensilla were observed, and on the tarsi, we documented trichoid sensilla resembling contact chemoreceptors. As in other taxa, no porous sensilla were noted.

Comparing all the studied taxa, the most visible feature is the presence of many different types of presumed mechanoreceptive sensilla. This diversity was also previously noted on the antennae of all nepomorphan taxa [41].

The other distinctive feature is the lack of porous sensilla on the legs of the studied specimens. This was expected, given the fact that olfactory sensilla are mostly present on the antennae and mouthparts of insects [26]. However, the sensilla placodea multilobated observed in Ochteridae and Gelastocoridae were considered to be possibly olfactory by Nowińska and Brożek [38]. Their presence on the legs might imply that they perform a different function. To draw an unquestionable conclusion, ultrastructural studies of these structures are required.

Apart from the putative mechanoreceptive and thermo-hygroreceptive sensilla noted in this study, other features were observed, which due to their lower relevance (because they are mostly species specific), were not taken into consideration while comparing the studied taxa.

Our study shows the presence of *pars stridens* (elements of stridulatory apparatus, in the form of pegs arranged in rows) in *Sigara* sp., on the forefemur of males and females (Figure 9). We describe it here separately from sensilla, as this element functions for species recognition and varies between species [4]. A stridulatory mechanism was also described for males of the *Nerthra* species in Gelastocoridae [43], but their legs are not involved in stridulation, which instead involves the genitalia. Stridulation in Corixidae was noted for the first time by Ball [44]. Jansson, in 1972 [45], studied sound production in males and females of *Cenocorixa blaisdelli* (Corixidae) and showed SEM microphotographs of the shape and location of their stridulatory apparatus.

Other cuticular structures were observed in Gelastocoridae (Figure 5c). There are short, flattened structures arranged in rows on the coxa of *G. flavus,* that cover a small surface near the connection with the trochanter. They resemble a stridulatory structure. However, the close proximity to longer setae might argue against it, as the setae might interfere with the stridulation. Additionally, as mentioned before, in the taxon Gelastocoridae only stridulation using genitalia has been described [46].

Apart from those structures, circular gaps in the hair pile with slightly raised edges were observed in Corixidae, distributed regularly between the hairs. They have not been described before and therefore their function is unknown. Given the fact that they were observed in aquatic insects, perhaps they might function as “suction cups” that improve grasping ability by trapping water or air, becoming adhesive. However more studies would be needed to fully understand their function.

Another interesting structure that appears in great numbers on the surface of the legs of Corixidae are flat, rounded fields with short, longitudinal indentations. To our knowledge they have not been observed until now, in any taxon, and therefore their function is unknown and only detailed future studies might propose their putative function.

Cuticular structures were also noted in the *Nerthra* species. We observed short-ribbed pegs on the edge of the femur (Figure 6b), which probably serve as auxiliary structures while grasping prey. In Gelastocoridae, this function is most likely performed by sensilla chaetica (SCh4), distributed in rows on the femur and trochanter.

Our studies on Ochteroidea fall in line with the newest phylogenetic studies [1]. The habitat transition, believed to have occurred in this group, is supported by the morphology of the legs as well as by the presence of sensilla similar to gustatory/tactile sensilla at the tip of the tarsus of terrestrial herbivorous insects. In the case of Corixidae, a clear conclusion regarding their position as a sister group to other nepomorphans [1] cannot be drawn because it requires broader studies, including other taxa than those dealt with here. Distinct differences between Corixidae and Ochteroidea were observed regarding the morphology of mechanoreceptive sensilla and lack of SPM and SS on the legs of Corixidae. However, most of these differences relate to the difference in habitats of these insects. Therefore, more studies on other Nepomorpha are needed to draw specific conclusions regarding the sensilla’s relevance in the taxa’s phylogeny.

## 5. Conclusions

Eight types of leg sensilla were observed in the studied nepomorphan taxa. While most of them are represented by a single type, sensilla trichodea (ST) were noted in the forms of six different subtypes, and sensilla chaetica consisted of four subtypes. The putative roles performed by the sensilla are mechanoreception (ST, SCh, SCa), contact chemoreception (SS), proprioception (SB) and thermo-hygroreception (SCo, SA). One type with an unknown function was observed (SPM).

Taking into account the results of the study, it is difficult to clearly confirm or reject the research hypothesis. The study does show that there are differences between the leg mechanoreceptive sensilla of Corixidae and Ochteroidea. In Corixidae, the specific types of sensilla were sensilla trichodea with fringes (ST5 and ST6), short sensilla chaetica present on the pala (SCh2) and sensilla chaetica present on the trochanter (SCh3). In Ochteroidea, two other types of mechanoreceptive trichoid sensilla (not observed in Corixidae) were present (ST3 and ST4), as well as structures most likely developed for grasping prey—ribbed cuticular pegs in the *Nerthra* species and sensilla chaetica in Gelastocoridae (SCh4). Therefore, a difference can be noted while comparing strictly aquatic and riparian Nepomorphans. However, all taxa display variations in the types of mechanoreceptive sensilla trichodea, which confirms the study of Nowińska and Brożek [41] stating that the whole infraorder of Nepomorpha can be distinguished by their large number of mechanoreceptive sensilla.

On the other hand, a difference between Ochteroidea and Corixidae can also be drawn by comparing sensilla other than mechanoreceptors. Sensilla observed only in Ochteroidea were ST1 on the end of the tarsus, which in herbivorous insects are regarded as gustatory/tactile sensilla. The sensillum styloconicum found in *Nerthra* might also perform a gustatory function. Another sensillar type noted only in Ochteroidea is SPM, which was also noted on the antennae of Gelastocoridae [38].

To conclude, it is possible to draw differences between strictly aquatic insects and riparian insects based on their leg sensilla. The most obvious difference is the presence of long, fringed, oar-like, presumably mechanoreceptive sensilla in the studied strictly aquatic taxa. However, a wide array of different sensilla, mostly mechanosensilla, were observed in all studied taxa of Nepomorpha. Therefore, it is difficult to confirm, that there is a notable difference in the number of types of sensilla between Corixidae and Ochteroidea.

There are thus some hints of differences in leg sensilla between the studied taxa. However, to further investigate the sensory adaptations of Nepomorpha, transmission electron microscopic and functional studies are required.

## Figures and Tables

**Figure 1 insects-14-00441-f001:**
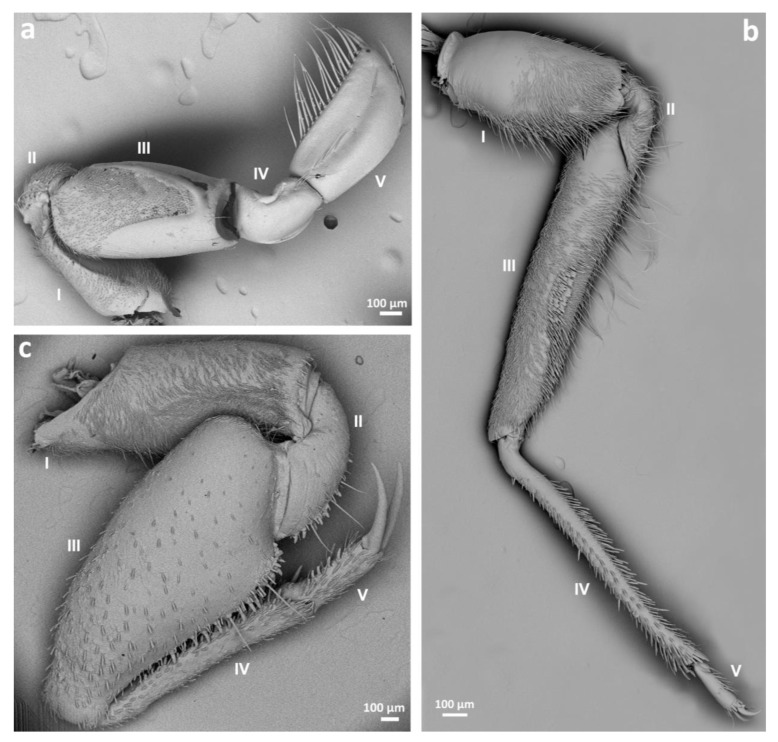
Morphology of the forelegs of *Sigara striata* (**a**); *Ochterus marginatus* (**b**); *Gelastocoris oculatus* (**c**). Segments of the forelegs: I—coxa, II—trochanter, III—femur, IV—tibia, V—tarsus.

**Figure 2 insects-14-00441-f002:**
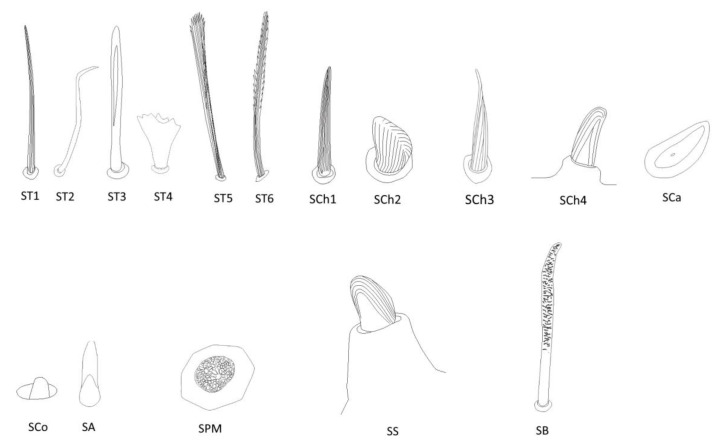
Types of sensilla in studied groups. ST—sensilla trichodea; SCh—sensilla chaetica; SCa—sensilla campaniformia; SCo—sensilla coeloconica; SA—sensilla ampullacea; SPM—sensilla placodea multilobated; SB—sensilla basiconica; SS—sensilla styloconica.

**Figure 3 insects-14-00441-f003:**
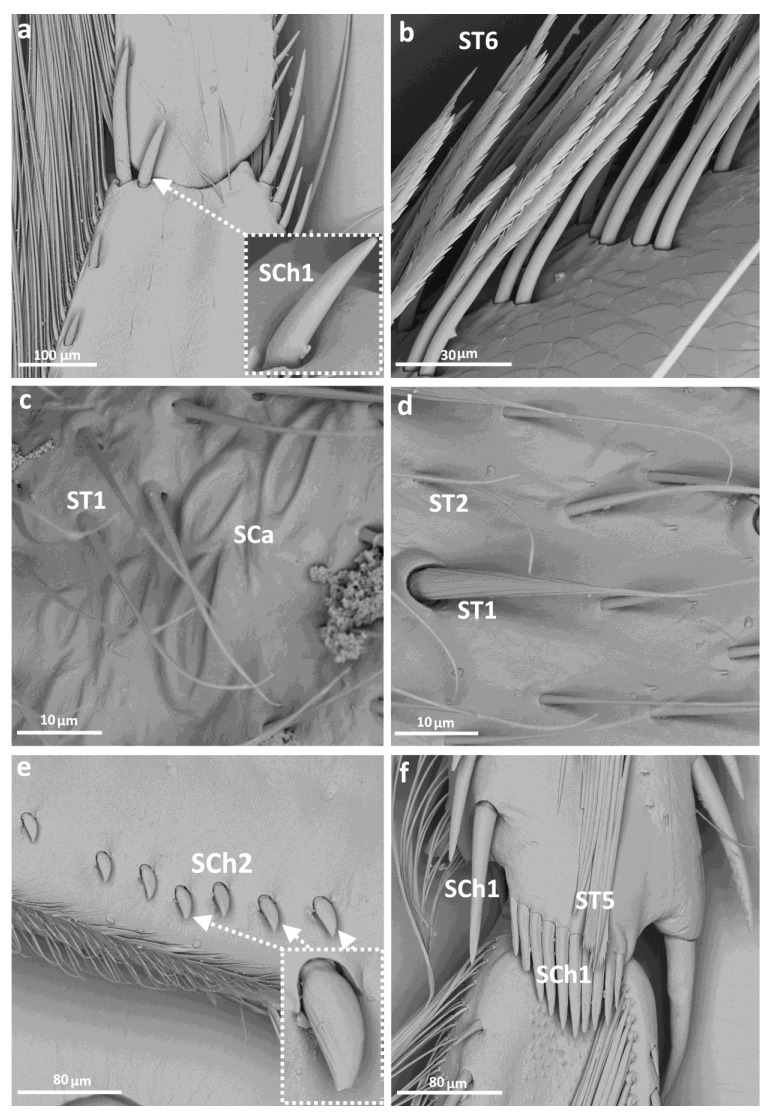
The leg sensilla of Corixidae: *Callicorixa preaeusta* (**a**,**b**); *Sigara falleni* (**c**); *Heliocorisa vermiculata* (**d**–**f**); SCa—sensilla campaniformia, SCh—sensilla chaetica, ST—sensilla trichodea.

**Figure 4 insects-14-00441-f004:**
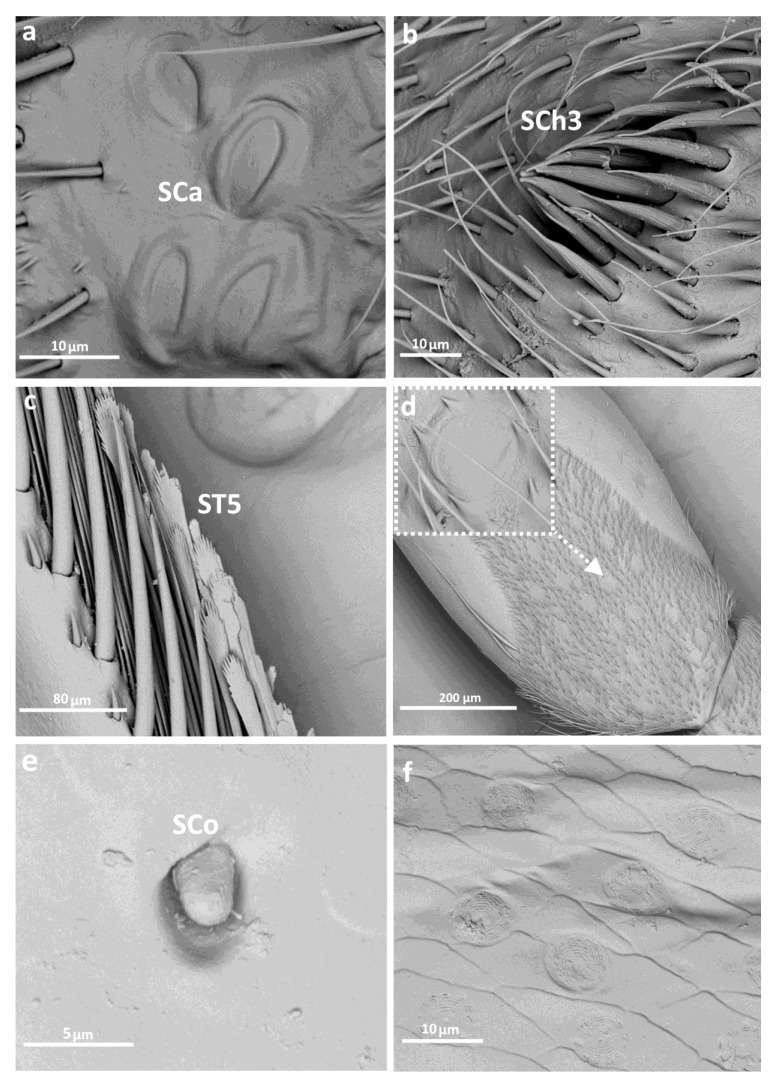
The leg sensilla of Corixidae: *Sigara falleni* (**a**–**c**); *S. striata* (**e**); *S.falleni* (**d**)—circular “gaps”; *S. striata* (**f**)—rounded fields; SCa—sensilla campaniformia, SCh—sensilla chaetica, ST—sensilla trichodea, SCo—sensilla coeloconica.

**Figure 5 insects-14-00441-f005:**
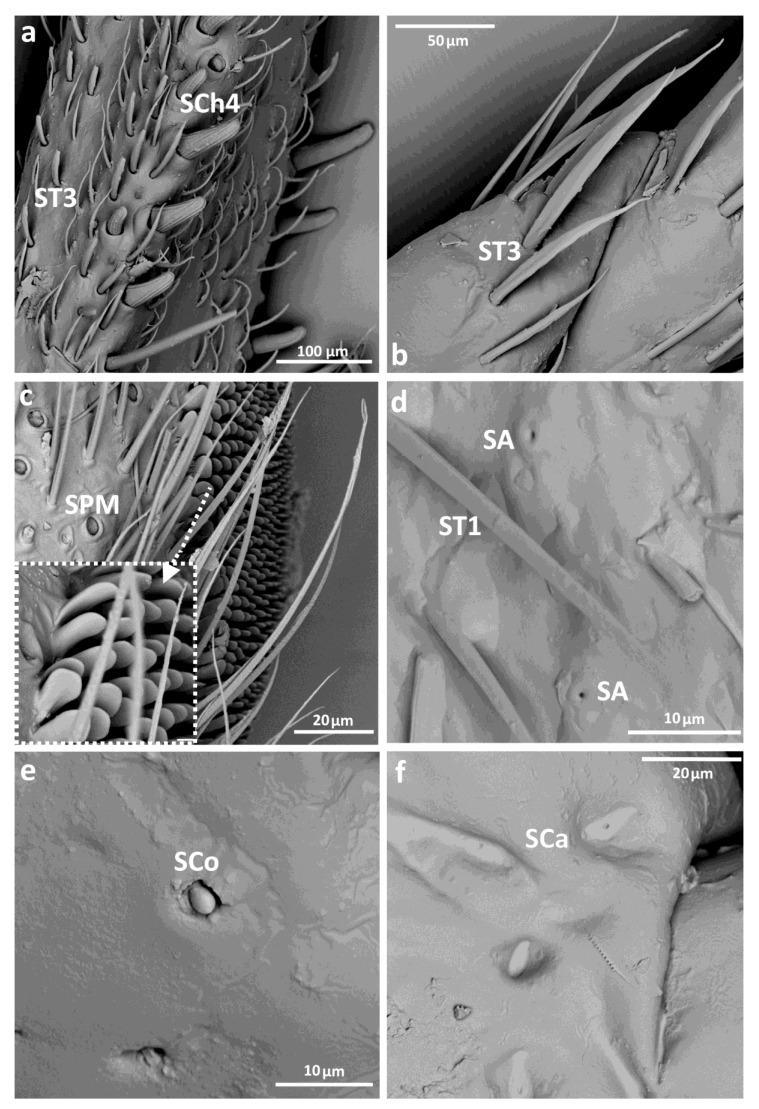
The leg sensilla of Gelastocoridae: *Gelastocoris flavus* (**a**–**c**,**e**,**f**); *G. oculatus* (**d**); SA—sensilla ampullacea, SCa—sensilla campaniformia, SCh—sensilla chaetica, SCo—sensilla coeloconica, SPM—sensilla placodea multilobated, ST—sensilla trichodea.

**Figure 6 insects-14-00441-f006:**
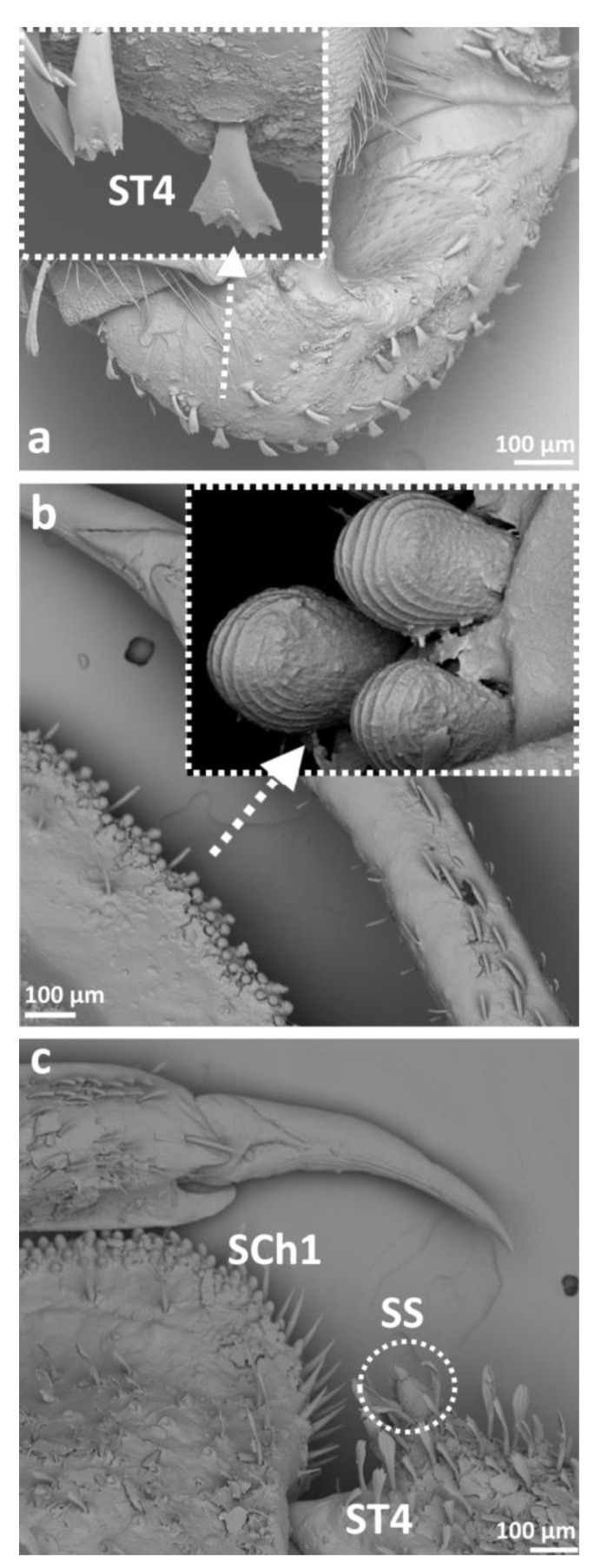
The leg sensilla of Gelastocoridae: *Nerthra grandicolis* (**a**); *N. mixta* (**c**); *N. colaticollis* (**b**)—ribbed pegs; SCh—sensilla chaetica, ST—sensilla trichodea, SS—sensilla styloconica.

**Figure 7 insects-14-00441-f007:**
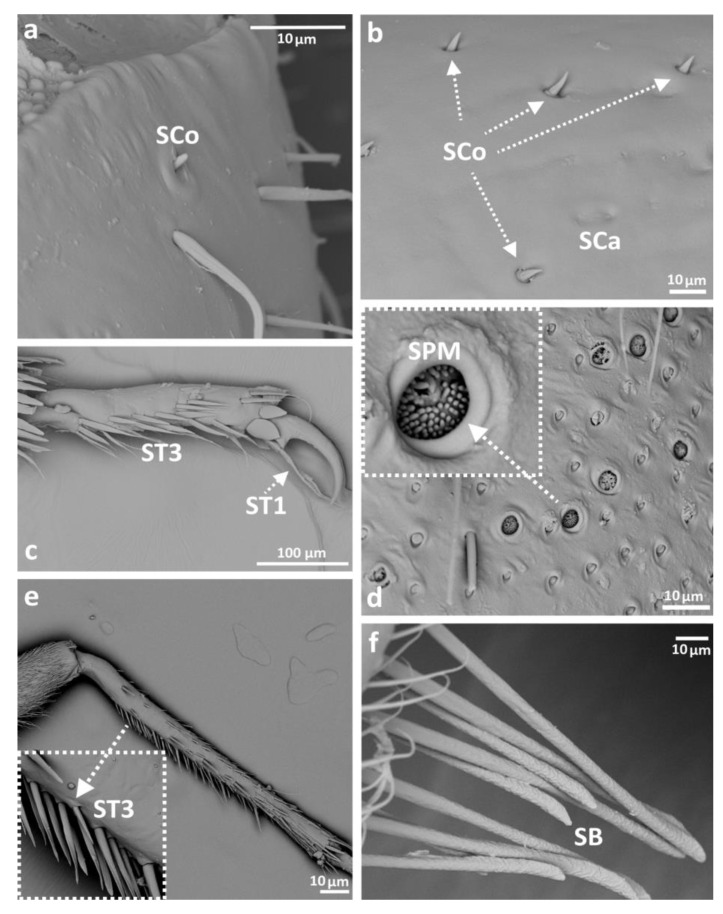
The leg sensilla of Ochteridae: *Ochterus marginatus* (**a**–**d**); *O. perbosci* (**e**,**f**); SB—sensilla basiconica, SCa—sensilla campaniformia, SCo—sensilla coeloconica, SPM—sensilla placodea multilobated, ST—sensilla trichodea.

**Figure 8 insects-14-00441-f008:**
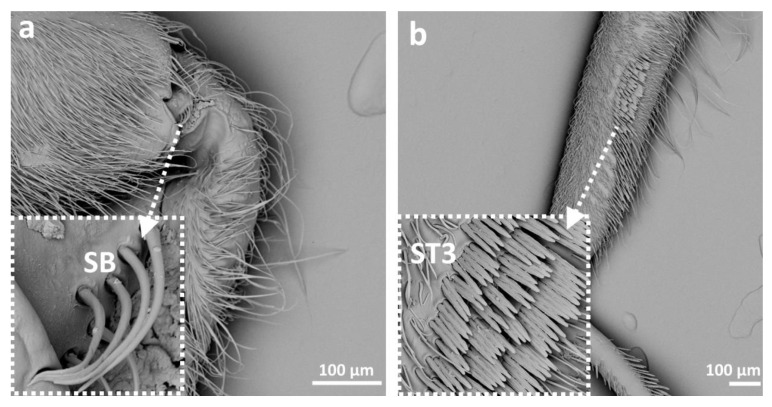
The leg sensilla of Ochteridae *Ochterus marginatus* (**a**,**b**); SB—sensilla basiconica, ST—sensilla trichodea.

**Figure 9 insects-14-00441-f009:**
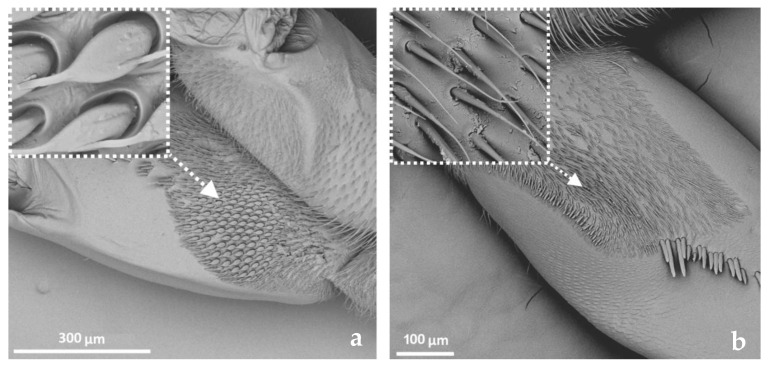
Pars stridens of *Sigara falleni*, male (**a**); female (**b**).

**Table 1 insects-14-00441-t001:** List of examined species.

Family	Genus	Species	Nr of Studied Specimens
Corixidae	*Callicorixa*	*Callicorixa preaeusta* (Fieber, 1848)	3
	*Heliocorisa*	*Heliocorisa vermiculata* (Puton, 1874)	2
	*Sigara*	*Sigara falleni* (Fieber, 1848) *Sigara striata* (Linnaeus, 1758)	3 4
Gelastocoridae	*Gelastocoris*	*Gelastocoris flavus* (Guérin-Méneville, 1835) *Gelastocoris oculatus* (Fabricius, 1798)	6 6
	*Nerthra*	*Nerthra colaticollis* (Todd, 1959) *Nerthra grandicollis* (Germar, 1837) *Nerthra mixta* (Montandon, 1929) *Nerthra ranina* (Herrich-Schäffer, 1853)	5 6 4 4
Ochteridae	*Ochterus*	*Ochterus marginatus* (Latreille, 1804) *Ochterus perbosci* (Guérin-Méneville, 1843)	8 5

**Table 2 insects-14-00441-t002:** List of observed sensilla in examined families.

Sensillum Type	Corixidae	Gelastocoridae	Ochteridae
ST1	+	+	+
ST2	+	+	+
ST3			+
ST4		+	
ST5	+		
ST6	+		
SCh1	+	+	
SCh2	+		
SCh3	+		
SCh4		+	
SCa	+	+	+
SCo	+	+	+
SA		+	
SB		+	+
SPM		+	+
SS		+	

"+" means the presence of all types of sensilla found in examined families.

## Data Availability

Materials obtained from the collections of the Moravian Museum in Brno, the Natural History Museum in Prague and the Naturalis Biodiversity Center in Leiden (Netherlands) are stored in the University of Silesia in Katowice, Institute of Biology, Biodiversity and Environmental Protection (Poland). The specimens obtained from the Naturalis Biodiversity Center in Leiden (Netherlands) are ultimately to be sent back and stored in the Naturalis Biodiversity Center.

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
