# Peer review of "The Leg Sensilla of Insects from Different Habitats—Comparison of Strictly Aquatic and Riparian Bugs (Corixidae, Ochteridae, Gelastocoridae: Nepomorpha: Insecta: Heteroptera)"

_insects, 2023, doi:10.3390/insects14050441_

Round 1

Reviewer 1 Report

As the authors state, the legs sensilla of Nepomorpha are an exciting topic, much more if they are investigated to compare sensory equipment related to different lifestyles with an evolutionary perspective.

This is the authors’ aim, but I think an SEM overview of legs sensilla, as the authors did, without any transmission electron microscopy observations, can be considered an excellent approach to pursue this goal. It is true that sensilla have been described in many insects and that their external morphology is related to their function. Still, this relationship is not unique, and internal morphology is necessary to explore the sensilla function in detail. This is particularly true in a group where correspondent sensilla (morphology and position) have never been described in detail with their internal morphology. 

In addition, the authors only observed two insects for each species and did not adequately describe their figures in the figure legends; this aspect must be improved to enhance clarity.

In the discussion, the authors state that Corixidae sensilla are different from the other studied taxa, but this is not supported by the results because, for example, they have two exclusive sensilla types while Gelastocorida has three exclusive sensilla. 

Later in the discussion, the authors state that “it is difficult to clearly confirm or reject the research hypothesis,” but in my opinion, it couldn’t be different with the research approach selected; more detailed research is needed to answer this question.

For these reasons, the paper could be published in its present form as a preliminary overview in a low-impact journal, but it needs to be improved to be published on Insects.  

Author Response

REVIEWER I

Dear Reviewer,

We thank you for all your insightful suggestions.
All the comments were taken into consideration.

Responds to the comments/suggestions are highlighted below (blue color).

We hope that after this revision our manuscript will be suitable for the Insects Journal.

Best regards,

Agnieszka Nowińska

Barbara Franielczyk-Pietyra

Dan A. Polhemus

As the authors state, the legs sensilla of Nepomorpha are an exciting topic, much more if they are investigated to compare sensory equipment related to different lifestyles with an evolutionary perspective.

This is the authors’ aim, but I think an SEM overview of legs sensilla, as the authors did, without any transmission electron microscopy observations, can be considered an excellent approach to pursue this goal. It is true that sensilla have been described in many insects and that their external morphology is related to their function. Still, this relationship is not unique, and internal morphology is necessary to explore the sensilla function in detail. This is particularly true in a group where correspondent sensilla (morphology and position) have never been described in detail with their internal morphology.

This manuscript provides a morphological background on nephomorphan`s leg sensilla. We wanted to find out what are the differences between the types and the number of leg`s sensilla among studied species. With this knowledge, we can conduct a project regarding the internal structure of some sensilla. With the material acquired so far (dry or stored in alcohol), we are not able to conduct a TEM study, which requires fresh specimens fixed in glutaraldehyde. We are, however, establishing cooperation with scientists who work in the field and would be able to acquire fresh specimens. Therefore, we are planning on expanding the study in the future.

In addition, the authors only observed two insects for each species and did not adequately describe their figures in the figure legends; this aspect must be improved to enhance clarity.

The figure legends were improved by adding explanations and now it should be more informative. Some elements, not considered as sensilla, were also described.

Regarding the number of studied specimens, in our methodology, we only point out the smallest amount used for this study (“At least 2 individuals of each species were examined for this study”). It does not mean that each species was studied on the basis of just two specimens. For example, many more were used for the Sigara species. As pointed out in the Materials and Methods, the material was obtained from international museums. Their approach to morphological study differs and the series of specimens available for the study is also different in each one. Therefore, some species were simply not available for us to obtain in bigger numbers, however, we did not want to exclude them from the study.

In the discussion, the authors state that Corixidae sensilla are different from the other studied taxa, but this is not supported by the results because, for example, they have two exclusive sensilla types while Gelastocorida has three exclusive sensilla.

There were some changes made in the results section, as thanks to the suggestions of another Reviewer, two more subtypes of sensilla chaetica specific to Corixidae were described. However, our approach to this comparison is based mostly on mechanoreceptive sensilla trichodea, since they are the most diverse. And there is a clear difference between the long, flattened, fringed sensilla in Corixidae and the much smoother, or shorter sensilla in Ochteroidea.

Later in the discussion, the authors state that “it is difficult to clearly confirm or reject the research hypothesis,” but in my opinion, it couldn’t be different with the research approach selected; more detailed research is needed to answer this question.

Thank you for your opinion. Our claim, that it is difficult to confirm or reject the research hypothesis, was based solely on the number of types of sensilla. We are confident that there are differences in the morphology of sensilla between these two taxa, however, given the fact that all of them possess great diversity in the shapes of sensilla, we could not clearly confirm that there is a notable difference in the number of types of sensilla. In our opinion, it makes Nepomorpha a very interesting and nuanced group.

For these reasons, the paper could be published in its present form as a preliminary overview in a low-impact journal, but it needs to be improved to be published on Insects.

The manuscript was reviewed according to the comments and changes were made. We hope that our manuscript might now be taken into consideration for publishing in Insects.

Reviewer 2 Report

The current article covers an essential systematic and environmental adaptive study of sensilla on forelegs in selected Nepomorpha taxa.

The paper meets the requirements of a scientific article and is well-prepared regarding the documentation presented. Only minor remarks were made on the distinction of sensilla cheatica, noting that the sensilla chaetic (SCh1) subtypes should be more clearly distinguished and described.

Minor remarks were posted in the discussion.

The detailed information there is in the PDF file and below: 1. The photos (Fig 3 a, e and f) of sensilla SCh1 and Sch1 in Fig 4 b represent different forms.

2. The shapes 3e and 3f is for me are different. Also, an essential difference in shape is between Sch1 and Sch1 in Fig 3e. Although they probably originate from the basal type of chaetic sensilla, several subtypes should be distinguished. In the discussion, It will be interesting to focus on and explain the variation of the sensilla in particular taxa of different habitats.

3. (line 319-322) - Such an assumption is misleading. It misrepresents the possibility of this sensillum in which the essential feature, i.e. the presence of a terminal pore, that indicates such function has yet to be demonstrated in this study. It is incomprehensible in this sense. It should be elaborated here that the discussions are based on specific examples, not just on dry citations [38]. Where the sensilla St1 was located, and what details morphological/physiological characters were analysed.

Author Response

REVIEWER II

Dear Reviewer,

We thank you for all your insightful suggestions and your support.
All general changes were done.

Responds to the comments/suggestions are highlighted below (blue color).

We hope that after this revision our manuscript will be suitable for the Insects Journal.

Best regards,

Agnieszka Nowińska

Barbara Franielczyk-Pietyra

Dan A. Polhemus

The current article covers an essential systematic and environmental adaptive study of sensilla on forelegs in selected Nepomorpha taxa.

The paper meets the requirements of a scientific article and is well-prepared regarding the documentation presented. Only minor remarks were made on the distinction of sensilla cheatica, noting that the sensilla chaetic (SCh1) subtypes should be more clearly distinguished and described.

Minor remarks were posted in the discussion.

The detailed information there is in the PDF file and below:

  1. The photos (Fig 3 a, e and f) of sensilla SCh1 and Sch1 in Fig 4 b represent different forms.

We agree with this statement. Therefore, the changes were made in the classification of the sensilla chaetica and two subtypes were added. In Fig. 3a and 3f, there is SCh1, while structure in fig. 3e was changed into SCh2 according to the reviewer’s suggestion (changes were made also in the Fig. 2 and Tab. 2). The sensillum in the Fig. 4b is now SCh3.

  1. The shapes 3e and 3f is for me are different. Also, an essential difference in shape is between Sch1 and Sch1 in Fig 3e. Although they probably originate from the basal type of chaetic sensilla, several subtypes should be distinguished. In the discussion, It will be interesting to focus on and explain the variation of the sensilla in particular taxa of different habitats.???

We distinguished 4 types of SCh: SCh1, SCh2, SCh3 and SCh4 (changes were made also in the Fig. 2 and Tab. 2). However, the type SCh2 occurs only in Corixidae, specifically on the pala and is used as a diagnostic feature for the species. Therefore, we do not hypothesize a specific role for them in this taxon.

  1. (line 319-322) - Such an assumption is misleading. It misrepresents the possibility of this sensillum in which the essential feature, i.e. the presence of a terminal pore, that indicates such function has yet to be demonstrated in this study. It is incomprehensible in this sense. It should be elaborated here that the discussions are based on specific examples, not just on dry citations [38]. Where the sensilla St1 was located, and what details morphological/physiological characters were analysed.

We would like to inform that the statement was further explained in the manuscript. We do not hypothesize the gustatory function of the sensillum, especially, when the taxon is not herbivorous. The comparison is strictly morphological and the presence of said sensillum is morphologically comparable with other terrestrial insects. Therefore, it was our goal to use it as another proof for our hypothesis regarding the taxon being similar to terrestrial insects.

Reviewer 3 Report

This paper presents the results of an investigation of the sensory structures on the legs of heteropteran insects belonging to three families. Members of one family, the Corixidae, live in aquatic habitats, whereas the others are riparian/terrestrial. Eleven species were studied, and their legs examined with SEM. The authors hypothesised that the leg sensilla of aquatic bugs would differ from those of the riparian species in both morphology and number. That is a reasonable hypothesis, but it would be helpful if reasons were provided as to why such differences might be expected. The paper is a descriptive one and is well written, although the discussion could be tidied up a little. It is profusely illustrated by excellent micrographs, which demonstrate the nature of the sensilla very well. The paper has an informative introduction, the methods are well described, and the relevant literature is cited appropriately. The simple summary and abstract at the beginning of the paper do their jobs well.

Results. The study found eight types of sensilla which are shown in the SEMs. Their basic structures are illustrated in the useful Figure 2. In addition, gross structure of the legs of species in the three families examined are described in some detail. This is the first paper to provide detailed descriptions of the leg sensilla of Nepomorpha, and in doing so makes a valuable contribution to knowledge of the group. My main concern is that the captions to figures need to provide more information on what they are and what they show without having to find that information somewhere in the text. Like tables, figures and their captions need to stand alone.

Specific comments

Line 16. “...Nepomorpha had not been studied until now.”

Line 44. Insert “in” after increase.

Line 46. By size do you mean length?

Line 83. “...branched or non-branched dendrites...”

Lines 94-95. Provide a rationale for the hypothesis (see comment above).

Line 104. How did you know there were no notable intersexual differences in sensilla types? Was this based on preliminary studies?

Table 1. I think the three columns should have headings (Family, Genus, Species), which would be in bold type, rather than Corixidae etc.

Line 137. Would triangular patch be a better term than surface?

Line 145. I am unsure what you mean by “connect with the tarsus”. Is this a physical connection or do the hairs interlock with those of the tarsus? Clarify.

Line 149. Gelastocoris should be in italics.

Line 173. Single not singular (and line 210)

Line 180. End not ending (and elsewhere)  

Line 226. “...and were generally most common...”

Line 238. “...first and second pairs of legs...”

Line 273. Basiconic sensilla (plural)

Line 280. What is meant by “largely” in, SPM were largely present... Clarify.

Table 2. Table 2 needs resetting so family names fit on a single line. The caption might be better to state “List of observed sensilla in examined families”.

Discussion. The discussion is not as well written as the rest of the paper and need some attention with respect to wording and sentence structure.

Line 296.I suggest “This paper presents the results of a morphological study of leg sensilla in 11 species in three families of Nepomorpha conducted with a scanning electron microscope”. Note:  only 11 species are listed in Table 1 (not 12).

Line 305. Adapted for walking

Line 306. Adapted for swimming

Line 314. “...were common to all....and are common on other body parts of these insects.

Line 316. Features

Line 318. Delete but

Line 334. “...trichodea have been found

Line 349. “...the presence of SPM had already been noted on the antennae.

Line 369. Insert “the” before presence.

Line 383. Has been described.

Line 399. What is the nature of the habitat transition? Spell it out in more detail.

Line 407. To draw specific conclusions about what? Clarify.

Line 429. Insert The before sensillum styloconicum

Line 430. “...Ochteroidea is the SPM, which was also noted...”

References. Check the format of references, in particular the use of capital letters in the titles of papers.

Line 513.Trichogramma minutum

The English is of a good standard although the discussion would benefit from some tidying up as indicated in my report.  

Author Response

REVIEWER III

Dear Reviewer,

We thank you for all your insightful suggestions and your support.
All general changes were done.

Responds to the comments/suggestions are highlighted below (blue color).

We hope that after this revision our manuscript will be suitable for Insects Journal.

Best regards,

Agnieszka Nowińska

Barbara Franielczyk-Pietyra

Dan A. Polhemus

This paper presents the results of an investigation of the sensory structures on the legs of heteropteran insects belonging to three families. Members of one family, the Corixidae, live in aquatic habitats, whereas the others are riparian/terrestrial. Eleven species were studied, and their legs examined with SEM. The authors hypothesised that the leg sensilla of aquatic bugs would differ from those of the riparian species in both morphology and number. That is a reasonable hypothesis, but it would be helpful if reasons were provided as to why such differences might be expected. The paper is a descriptive one and is well written, although the discussion could be tidied up a little. It is profusely illustrated by excellent micrographs, which demonstrate the nature of the sensilla very well. The paper has an informative introduction, the methods are well described, and the relevant literature is cited appropriately. The simple summary and abstract at the beginning of the paper do their jobs well.

Results. The study found eight types of sensilla which are shown in the SEMs. Their basic structures are illustrated in the useful Figure 2. In addition, gross structure of the legs of species in the three families examined are described in some detail. This is the first paper to provide detailed descriptions of the leg sensilla of Nepomorpha, and in doing so makes a valuable contribution to knowledge of the group. My main concern is that the captions to figures need to provide more information on what they are and what they show without having to find that information somewhere in the text. Like tables, figures and their captions need to stand alone.

Captions to figures were improved by adding explanations and now it should be more informative.

Specific comments – most of the changes were made and for some comments, we added explanations.

Line 16. “...Nepomorpha had not been studied until now.” Corrected

Line 44. Insert “in” after increase. Corrected

Line 46. By size do you mean length? Yes, we corrected it in the text.

Line 83. “...branched or non-branched dendrites...” Corrected

Lines 94-95. Provide a rationale for the hypothesis (see comment above). The reasoning behind the hypothesis was based on apparent differences between the antennal sensilla of aquatic and terrestrial insects. Therefore, we hypothesised that leg sensilla will also differ due to the different environment. The explanation was added to the manuscript.

Line 104. How did you know there were no notable intersexual differences in sensilla types? Was this based on preliminary studies? Yes, it was. We added this information to the manuscript.

Table 1. I think the three columns should have headings (Family, Genus, Species), which would be in bold type, rather than Corixidae etc. Yes, it was a mistake during the editing. Changes were made according to the suggestion.

Line 137. Would triangular patch be a better term than surface? We agree, it was changed in the manuscript.

Line 145. I am unsure what you mean by “connect with the tarsus”. Is this a physical connection or do the hairs interlock with those of the tarsus? Clarify. No, these hairs are very long and reach the tarsus, there is no interlock. We corrected that in the manuscript.

Line 149. Gelastocoris should be in italics. Corrected.

Line 173. Single not singular (and line 210) Corrected.

Line 180. End not ending (and elsewhere) Corrected.

Line 226. “...and were generally most common...” Corrected.

Line 238. “...first and second pairs of legs...” Corrected.

Line 273. Basiconic sensilla (plural) Corrected.

Line 280. What is meant by “largely” in, SPM were largely present... Clarify. We meant that they were present in a big amount.

Table 2. Table 2 needs resetting so family names fit on a single line. The caption might be better to state “List of observed sensilla in examined families”. Changed.

Discussion. The discussion is not as well written as the rest of the paper and need some attention with respect to wording and sentence structure.

Line 296.I suggest “This paper presents the results of a morphological study of leg sensilla in 11 species in three families of Nepomorpha conducted with a scanning electron microscope”. Corrected.

Note:  only 11 species are listed in Table 1 (not 12). There are 12 species, probably the bold type of Corixidae/Callicorixa etc. was misleading.

Line 305. Adapted for walking Corrected.

Line 306. Adapted for swimming Corrected.

Line 314. “...were common to all....and are common on other body parts of these insects. Corrected.

Line 316. Features Corrected.

Line 318. Delete but Corrected.

Line 334. “...trichodea have been found Corrected.

Line 349. “...the presence of SPM had already been noted on the antennae. Corrected.

Line 369. Insert “the” before presence. Corrected.

Line 383. Has been described. Corrected.

Line 399. What is the nature of the habitat transition? Spell it out in more detail. This part was explained in the text but in the introduction part – please see lines 52-54.

Line 407. To draw specific conclusions about what? Clarify. Regarding the sensilla relevance in the taxa phylogeny. It was corrected in the text.

Line 429. Insert The before sensillum styloconicum Corrected.

Line 430. “...Ochteroidea is the SPM, which was also noted...” Corrected.

References. Check the format of references, in particular the use of capital letters in the titles of papers. Corrected.

Line 513.Trichogramma minutum Corrected.

The English is of a good standard although the discussion would benefit from some tidying up as indicated in my report.

The English was checked again by the native speaker and the discussion was improved.

Round 2

Reviewer 1 Report

I appreciate the effort of the authors to improve the paper. Still, for the same reason I already expressed in my previous evaluation (discrepancy between the aim of the paper and the methods), in my opinion, the article in its present form, as a preliminary overview, could be published in a lower-impact journal, but it needs to be improved to be published on Insects.

Author Response

Dear Reviewer,

We improved our paper once more.

The aim of the study was rewritten to better describe that only the morphology of the legs was studied and that it is a first attempt in describing leg sensory structures in Nepomorpha.

A disclaimer to conclusions was added that despite some hints of differences of leg sensilla are visible, TEM studies are required to further investigate the subject.

Thank you for you concerns regarding the ultrastructure of studied sensilla.

We made it a point to clearly state in the manuscript that all the functions of described sensilla are solely presumed.

Best regards,

A. Nowińska, B. Franielczyk-Pietyra, A.D. Polhemus